# Physical, Thermal Transport, and Compressive Properties of Epoxy Composite Filled with Graphitic- and Ceramic-Based Thermally Conductive Nanofillers

**DOI:** 10.3390/polym14051014

**Published:** 2022-03-03

**Authors:** Siti Salmi Samsudin, Mohd Shukry Abdul Majid, Mohd Ridzuan Mohd Jamir, Azlin Fazlina Osman, Mariatti Jaafar, Hassan A. Alshahrani

**Affiliations:** 1Kampus Tetap Pauh Putra, Faculty of Mechanical Engineering Technology, Universiti Malaysia Perlis (UniMAP), Arau 026000, Perlis, Malaysia; sitisalmi@unimap.edu.my (S.S.S.); ridzuanjamir@unimap.edu.my (M.R.M.J.); 2Faculty of Chemical Engineering Technology, Universiti Malaysia Perlis (UniMAP), Arau 02600, Perlis, Malaysia; azlin@unimap.edu.my; 3School of Materials and Mineral Resources, Universiti Sains Malaysia, Nibong Tebal 14300, Pulau Pinang, Malaysia; mariatti@usm.my; 4Department of Mechanical Engineering, College of Engineering, Najran University, Najran 11001, Saudi Arabia; haalshahrani@nu.edu.sa

**Keywords:** thermal properties, compressive properties, nanofilled composites, thermally conductive, nanocomposites

## Abstract

Epoxy polymer composites embedded with thermally conductive nanofillers play an important role in the thermal management of polymer microelectronic packages, since they can provide thermal conduction properties with electrically insulating properties. An epoxy composite system filled with graphitic-based fillers; multi-walled carbon nanotubes (MWCNTs), graphene nanoplatelets (GNPs) and ceramic-based filler; silicon carbide nanoparticles (SiCs) was investigated as a form of thermal-effective reinforcement for epoxy matrices. The epoxy composites were fabricated using a simple fabrication method, which included ultrasonication and planetary centrifugal mixing. The effect of graphite-based and ceramic-based fillers on the thermal conductivity was measured by the transient plane source method, while the glass transition temperature of the fully cured samples was studied by differential scanning calorimetry. Thermal gravimetric analysis was adopted to study the thermal stability of the samples, and the compressive properties of different filler loadings (1–5 vol.%) were also discussed. The glass temperatures and thermal stabilities of the epoxy system were increased when incorporated with the graphite- and ceramic-based fillers. These results can be correlated with the thermal conductivity of the samples, which was found to increase with the increase in the filler loadings, except for the epoxy/SiCs composites. The thermal conductivity of the composites increased to 0.4 W/mK with 5 vol.% of MWCNTs, which is a 100% improvement over pure epoxy. The GNPs, SiCs, and MWCNTs showed uniform dispersion in the epoxy matrix and well-established thermally conductive pathways.

## 1. Introduction

Epoxy resin is preferred as the ideal thermal management material (TMM) for packaging in the power and electronic industries because of its superior electrical insulation, corrosion resistance, adhesion, easy processing ability, light weight, and low cost [1]. However, the low thermal conductivity of pure epoxy (0.18–0.22 W/mK) has become a major drawback in heat dissipation. Researchers working in the field of electronics face a major challenge in resolving this issue [2,3]. Some researchers have made several attempts to improve the thermal conductivity of epoxy by incorporating fillers with high thermal conductivity, such as graphite- and ceramic-based nanofillers [4,5,6]. Nevertheless, significant filler loading is frequently necessary to improve thermal conductivity, weakening the polymer’s exceptional mechanical properties and processing characteristics. As a result, it is critical to develop high thermal conductivity in polymer composites with minimal filler loading.

High-aspect-ratio nanoparticles have received particular attention in the development of an effective filler for this purpose, since they can effortlessly demonstrate thermally conductive pathways in polymer composites [7]. These nanoparticles include graphitic-based fillers, carbon nanotubes (CNTs), and graphene nanoplatelets (GNPs). These graphitic-based fillers have been proved as a good choice due to their ultrahigh thermal conductivity and large aspect ratio [5]. In particular, CNTs have a very high thermal conductivity value (~3000 W/mk for individual CNT [8]), while that of GNPs is ~5000 W/mK [9], thus making them excellent fillers for polymer composites [10,11,12]. Unfortunately, instead of creating tight filler-to-filler junctions, they have a propensity to form filler-to-filler junctions that spread phonons, increasing local thermal resistance, resulting in low thermal conductivity values for composite laminates [8,12]. Therefore, perfectly preparing these graphitic-based fillers for successful use in polymer composites is quite challenging.

On the other hand, ceramic-based fillers, such as aluminium nitride, aluminium oxide, boron nitride, and silicon carbide (SiC), have long been regarded as highly thermally conductive ceramic particles with good electrical insulating properties [13,14]. The primary mechanism is to enhance the formation of thermally conductive pathways, chains, and networks in the matrix by adding fillers with high conductivity coefficients. For practical applications in the electrical and electronic fields, SiCs have gained considerable attention because of their excellent properties, such as high thermal conductivity (~390 W/mK), low thermal expansion coefficients, high breakdown fields, and excellent mechanical properties, as well as their inertness towards chemicals, electrical insulation, low cost and abundant supply [15]. Thus, they can be applied in different conditions, such as under high electron density, frequency, and temperatures, and in harsh environments. However, SiCs are difficult to disperse in polymer matrices due to the high surface area, strong van der Waals interactions, and hydrophobicity. Despite the surface modifications that have been implemented to improve the characteristics of SiCs, constructing extremely effective thermally conductive pathways in composites with minimal filler loading remains a challenge [14].

Many studies have shown that adding graphite-based, thermally conductive fillers to the polymer matrix improves thermal conductivity [16,17,18], but only a few have shown that adding ceramic-based thermally conductive fillers improves thermal conductivity. Furthermore, no systematic report on silicon carbide nanoparticles as filler materials for thermally conductive polymer composites has been reported. Most studies on silicon carbide were limited to silicon carbide microparticles and whiskers [19,20,21,22]. Furthermore, there is less comparison between graphite-based and ceramic-based thermally conductive nanofillers published in single articles.

The dispersion of fillers in the matrix material is one of the main difficulties in the manufacturing process of particulate composites. However, well-distributed nanofillers can lead to new and unique composite properties, such as enhanced stiffness and toughness, with minimal filler loading [23]. Dissolvers and bead mills are among the most common means used for dispersing particle agglomerates in liquids [24]. Dispersion aided by ultrasonic waves is another option. The mechanical properties of the matrix may be influenced by ultrasonic treatment [25].

The novel aspect of this work is the utilisation of methyl ethyl ketone (MEK) as the solvent for the solution mixing method. This simple method was implemented as a form of double dispersion; MEK was used to premix the filler in the ultrasonic first, and later mixed again in planetary centrifugal mixing with the presence of epoxy and the absence of air. Because most of the solvent (MEK) could be removed during the mixing process, this approach is different from others, which require an additional step to completely remove the used solvent after the mixing process [26,27]. Additionally, the pressure utilised in this work was used to eliminate the unnecessary remaining composite mixture.

In this study, the effect of three different nanofillers, MWCNTs, GNPs, and SiCs, on the physical, thermal transport, and compressive properties of epoxy nanocomposites was investigated. Therefore, neat and reinforced nanocomposite bulk samples were manufactured using a simple approach of processing and further compressing them during hardening. Next, the neat and epoxy nanocomposites were investigated for their microstructures and physical, mechanical, and thermal conductivity. The compression strengths of the neat and epoxy nanocomposites were experimentally obtained and compared. Moreover, the fractured surfaces of the composites were analysed using scanning electron microscopy (SEM) fractography.

This paper is a preliminary part of a project on developing heat-conductive composites with electrical insulators that are reinforced with nanofillers for high-end applications, such as thermal management materials (TMMs). An ideal TMM must have excellent thermal conductivity to address heat dissipation issues. It must be able to withstand harsh environments, such as high temperature and high humidity, over a long period, without the degradation of its thermal conductivity. Generally, these environments require the materials to have very high thermal conductivity, around >2.5 W/mK. High thermal conductivity, high electrical resistivity, low coefficient of thermal expansion, and non-toxicity are the main reasons why these three materials were chosen as the nanofiller embedded into the epoxy matrix. Although the thermal conductivity values were far short of those at which we aimed, the materials are still under development.

## 2. Experimental Method

### 2.1. Materials

GNPs 0540DX and SiCs 6820HK were provided by SkySpring Nanomaterials Inc. (Houston, TX, USA). Hydroxyl-functionalised MWCNTs UCNT-COOH were provided by United Nanotech Innovations Pvt. (Bangalore, India). The GNP and SiC nanoparticles were used as obtained, without any further functionalisation. An epoxy resin called EpoxAmite 100 was used as a matrix in this study, and the selected curing agent was EpoxAmite 103 Slow. Both chemical products were manufactured by the Smooth-On Inc. (Lehigh, PA, USA). and supplied by Castmech Technologies Sdn. Bhd (Perak, Malaysia). The densities of the epoxy resin and the curing agent were 1.10 and 0.96 g/cm^3^, respectively. The weight ratio of epoxy resin to curing agent was 4:1, as suggested by the manufacturer. Methyl ethyl ketone (MEK) was purchased from Merck Sdn. Bhd (Selangor, Malaysia) and used as a dispersion solvent. The properties of the fillers utilised in this study are listed in Table 1.

### 2.2. Sample Preparation

The as-received nanoparticles were dispersed in a solution comprising MEK and epoxy resin through ultrasonication at 40% amplitude for 45 min to break any agglomerations between the nanoparticles. The mixture’s temperature was always maintained below 20 °C using an ice bath to avoid an increase in temperature during the sonication process. After the completion of the dispersion stage, the suspension comprising epoxy, MEK, and nanoparticles was mixed via a planetary centrifugal mixer (Thinky ARE 310, Thinky Corp., Tokyo, Japan), at a rotation speed of 2000 rpm for 5 min and a revolution speed of 1000 rpm for 1 min to disperse the particles in the resin. Next, the curing agent was added to the obtained dispersion’s mixture, and the mixture was stirred for 2 min at a rotation speed of 2000 rpm and then for the 30s at a rotation speed of 1000 rpm. As most of the solvent (MEK) can be removed during the mixing process, this method differs from others for making filler-based composites because it ensures good dispersion of fillers as the solvent level decreases [28]. After the completion of mixing in the planetary centrifugal mixer, the uncured epoxy/nanofiller mixture was cast into a mould and subjected to a pressure of 40 MPa for 2 h. Next, the pressure was released to remove any remaining composite mixture. The epoxy nanocomposites were cured for 24 h at room temperature before being heated to 93 °C in an oven for an additional hour. After allowing the epoxy nanocomposites to cool to room temperature, the samples were removed from the mould to be physically analysed. This sample preparation method has the advantage of decreasing the number of air voids while using high pressure during the curing process (planetary centrifugal mixer), which is critical for obtaining high-quality composite samples with fewer air voids and less porosity [28]. Figure 1 depicts the composite samples’ illustration preparation process. First, the epoxy nanocomposites were fabricated using filler loadings of MWCNT, GNP, and SiC contents (1–5 vol.%). The samples were as follows: pure epoxy for the pure epoxy without any fillers, epoxy/MWCNTs 1–5 for the epoxy composites filled with MWCNT, epoxy/GNPs 1–5 for the epoxy composites filled with GNP, and, lastly, epoxy/SiCs 1–5 for the epoxy composites filled with SiC fillers.

### 2.3. Evaluation Method

#### 2.3.1. Density Measurements

The sample densities were determined using Archimedes’ principle and an electronic scale (Mettler Toledo). The samples were first weighed while dry (*W_dry_*) and then weighed after submerging in water, (*W_wet_*). From these data, the density *(ρ)* of the sample was calculated using Equation (1). Further, the theoretical density was calculated for each composite formulation using Equation (2). In these equations, *ϕ_i_* is used for the weight fraction of the constituents; ‘*ρ_i_*’ is the density of the constituents, and ‘*ρ_Theo_*’ is the calculated theoretical density.
(1)$$\rho =\frac{W_{dry}}{W_{dry}-W_{wet}}\times \rho_{water}\left(T\right)$$
(2)$$\rho_{Theo}=\frac{1}{\sum_{i}\frac{\phi_{i}}{\rho_{i}}}$$

#### 2.3.2. Thermal Conductivity

To measure thermal conductivity, three samples in a cylindrical form with a diameter of 20 mm and a thickness of 5 mm were prepared for each composition. To meet the required thermal contact with the sensor, emery paper (120, 400, and 600 grit size, in sequence) were used to hand-polish the samples. Thermal conductivity of both the pure epoxy and epoxy nanocomposites was measured using Hot Disk TPS 2500s (Gothenburg, Sweden), a hot disc thermal analyser, through the transient plane source (TPS) method. In the TPS method, an electrically insulated flat nickel sensor was placed between two substrate pieces. The sensor served as a heater and a thermometer at the same time. During the measurement, a current pulse was passed through the sensor to generate a heatwave. The rise in temperature as a function of time was used to determine the material’s thermal properties. The time and input power were chosen so that the heat flow was contained within the sample boundaries, and the temperature rise of the sensor was unaffected by the sample’s outer boundaries. A series of four measurements was recorded in each sample with a suitable time interval between the measurements. This time interval allowed the sample to revert to its initial state. Typically, the difference between these measurements was 0.05 W/mK.

#### 2.3.3. Differential Scanning Calorimetry (DSC) Analysis

The glass transition temperature (*T_g_*) of the pure epoxy and epoxy nanocomposites was measured using the Perkin Elmer DSC 8000 (Perkin Elmer, Waltham, MA, USA) instrument. The measurements were performed in a dry atmosphere, a temperature range of 50 °C to 250 °C, and a heating rate of 10 °C/min under a nitrogen atmosphere. The nitrogen flow rate was kept constant at 50 mL/min.

#### 2.3.4. Thermal Gravimetric Analysis (TGA)

TGA5500, TA Instruments (New Castle, DE, USA), was used to analyse dynamic thermogravimetry from 25 °C to 800 °C. During the analysis, nitrogen flow was kept at a constant rate of 50 mL/min and a heating rate of 10 °C/min. The sample was weighed at 10 mg and placed in a 100 mL crucible according to the ASTM E1131 procedure.

#### 2.3.5. Compression Properties

The compression performance of the epoxy nanocomposites was determined through a compression test using Instron-3369 (Instron, Waltham, MA, USA), a universal testing machine. The samples used were in the form of a cylinder with a diameter of 12.7 mm and 5 mm thickness as per the ASTM D 695-15 standard. The compression rate was maintained at 1 mm/min throughout the test. The compression strength and modulus of the composites were determined using five replicates each, and the average values were reported.

#### 2.3.6. Scanning Electron Microscopy (SEM)

The morphologies of the fractured surfaces of the epoxy nanocomposites were characterised using SEM TESCAN VEGA 4 (TESCAN VEGA, Brno-Kohoutovice, Czech Republic). The fractured parts of the samples were trimmed before scanning, and gold was properly coated throughout the samples’ surfaces. SEM images were obtained at accelerating voltages ranging from 5 to 15 kV and magnifications ranging from 100× to 1000×.

## 3. Results and Discussion

### 3.1. Density Measurements

Figure 2 shows the effect of the MWCNT, GNP, and SiCs loading (vol.%) on the density of the epoxy nanocomposite system, which is an excellent way to detect voids in cured samples. If the fillers were evenly distributed and dispersed throughout the matrix, the density of the nanocomposites increased with increasing filler loading. Increasing the filler loading from 0 vol.% to 5 vol.% for the epoxy filled with graphite-based nanofillers and up to 4 vol.% for epoxy/SiCs increased the density of the nanocomposite system. However, the voids and holes between the filler and the matrix in the composites caused by the aggregations of the filler at high-filler loading decreased the density of the composites. The epoxy nanocomposites filled with graphite-based fillers (MWCNTs and GNPs) had a denser structure with the incorporation of 5 vol.% filler loading compared with that of epoxy alone. Therefore, the density of the epoxy nanocomposites filled with graphite-based fillers (MWCNTs and GNPs) increased compared with the pure epoxy. As a result, when the amount of nanofillers increased, the space between the polymer chains became filled to a considerable degree with nanofillers instead of being empty, as in the pure epoxy resins. Consequently, the overall density of the epoxy nanocomposite system increased owing to the compact filling of the nanofillers between the polymer chains at a fixed epoxy matrix volume.

However, the agglomeration occurred at higher SiC loading and created voids and stress concentration sites that resulted in the failure of the nanocomposites. The epoxy/SiCs 5′s density decreases might have been due to the void formation in the samples. If more SiCs were added to the epoxy nanocomposite mixture, the viscosity of the epoxy nanocomposite mixture would increase dramatically, and it would turn into a stiff paste at around 5 vol.%. More than 5 vol.% of SiC content could not be achieved for the SiC particles because, for the same SiC content, the surface-area-to-volume ratio increased as the particle size decreased because of the high aspect ratio, resulting in the epoxy matrix failing to bind the nanofillers. This could have been attributed to the filler sedimentation in the suspension and unfilled spaces in the samples themselves [29]. With such high filler contents, it is impossible to degas the samples fully; therefore, some air bubbles will likely remain, reducing the density and thermal conductivity. As the uncured mixture was a very stiff paste, the degassing procedure failed to remove the air bubbles formed during the mixing process. Therefore, this composite could have a considerable void content.

### 3.2. Thermal Properties

The thermal conductivity and diffusivity of various epoxy nanocomposite systems containing MWCNT, GNP, and SiC nanofillers were examined, and the results are presented in Figure 3. With its low thermal conductivity (~0.2 W/mK), epoxy polymer frequently requires the addition of a thermally conductive filler to achieve an acceptable level of heat transfer. Figure 3a shows that the epoxy matrix filled with all the nanofillers demonstrated higher thermal conductivity compared with that of the pure epoxy. Additionally, the thermal conductivity of the epoxy nanocomposites increased gradually as the content of the MWCNT and GNP nanofillers increased, with the exception of the SiC nanofillers, in which the thermal conductivity value decreased somehow as the filler loading increased to 5 vol.%. The incremental trends of thermal conductivity via the addition of MWCNTs and GNPs in epoxy nanocomposites have been reported by other researchers in previous studies [6,30,31,32]. The thermal conductivity of epoxy nanocomposites shows only a slight improvement in the thermal conductivity value at low filler loadings (up to 1 vol.%). These fillers are in an isolated state, akin to a ‘sea-island’ structure; hence, the amount of nanofillers required to construct a continuous network would be insufficient. A higher concentration of nanofillers (3–5 vol.%) results in a faster increase in thermal conductivity, except for SiC nanofillers.

Figure 3a shows that the epoxy/MWCNTs nanocomposites demonstrated the highest thermal conductivity value for each filler loading, followed by the epoxy/GNPs and epoxy/SiCs nanocomposites. The amount of MWCNTs and GNPs in a nanocomposite enhances its heat conductivity exponentially. The nanofillers’ thermal conductivity determines the nanocomposites’ thermal conductivity. As mentioned above, MWCNT is a graphitic-based nanofiller with exceptionally high thermal conductivity (up to approximately 3000 W/mK) [33]. Thus, regardless of how low the filling content of MWCNT was, the nanocomposite’s thermal conductivity did not reduce due to the interfacial thermal resistance. A 3D thermal conductivity network formed inside the epoxy nanocomposite when the MWCNT content reached 3 vol.%, increasing the thermal conductivity up to 0.32 W/mK. Compared with epoxy/GNPs 3 and epoxy/SiCs 3 in the same concentration, there was only a slight increase in thermal conductivity, 0.30 W/mK and 0.25 W/mK, respectively. Therefore, the filler’s dispersion at relatively high contents (>3 vol.%) had a crucial impact on the nanocomposite’s thermal conductivity. As the MWCNTs had a higher aspect ratio and better dispersion in the epoxy matrix than the GNPs, the acoustic phonons could travel along the nanotube pathways much faster than the graphene.

It was discovered that the epoxy matrix had a thermal bridge created by the MWCNTs and GNPs, which improved heat transfer. As a result, the thermal chains created between the fillers improved the epoxy nanocomposites’ thermal conductivity. Both epoxy/MWCNTs and epoxy/GNPs presented an increase in thermal conductivity because MWCNTs and GNPs have extremely high thermal conductivities (up to 3000 W/mK [33] and 5000 W/mK [30], respectively). With increasing MWCNT and GNP filler loading, the thermal conductivity of the epoxy/MWCNTs and epoxy/GNPs increased simultaneously. Due to the shorter distance between the fillers, more thermal chains could form, establishing a thermal conducting channel or network that conducted heat. At 1 vol.% of filler loading, the thermal conductivity of the epoxy nanocomposites containing MWCNTs, GNPs, and SiCs increased from 0.20 W/mK to 0.26 W/mK, 0.25 W/mK, and 0.24 W/mK, respectively. As the MWCNT and GNP concentration increased to 5 vol.%, the epoxy/MWCNTs still showed better thermal conductivity than the epoxy/GNPs and epoxy/SiCs. The epoxy/MWCNT 5 showed a thermal conductivity of 0.4 W/mK, with an increment of ~100% compared to the pure epoxy.

The epoxy/GNP 5 showed a thermal conductivity of 0.36 W/mK, with an increase of 80% compared to the pure epoxy. Lastly, the epoxy/SiCs 5 presented the lowest enhancement, only 40%, with a 0.28 W/mK. Therefore, the MWCNTs employed in this study had better thermal conductivity than those of the GNPs and SiCs, as the MWCNTs had lower filler-to-matrix interface resistance owing to the COOH-functionalisation. In addition, increasing the filler loading from 4 to 5 vol.% resulted in a substantial rise in thermal conductivity. It is speculated that the percolation threshold was obtained at ~4 vol.%, and that a continuous MWCNT network was formed. The greater aspect ratio of MWCNTs compared with those of GNPs and SiCs and the superior interaction between MWCNTs and epoxy matrix could all be contributing factors for a well-bridged and effective thermal conduction network between nanotubes [34]. Phonons play a critical role in the heat conduction of most solid materials, as is well known. Semiconductors’ thermal conductivity is determined by two factors: the interaction of harmonic or anharmonic phonons at high temperatures and phonon scattering, at low temperatures, by crystal boundaries [34].

At the same filling amount of 5 vol.%, the thermal conductivity of the epoxy/GNP 5 was higher than that of the epoxy/SiC 5 because the GNPs had a larger aspect ratio than the SiCs, which are more conducive to the formation of thermal conductivity networks. Because SiCs have a larger surface area, agglomerations formed in the composites at this filler loading, preventing an efficient thermal conduction network in the epoxy. This resulted in low thermal conductivity values for the epoxy/SiCs 5 composites compared with those of the epoxy/MWCNT5 and epoxy/GNP 5 composites. Substantial thermal interface resistance was caused by a phonon mismatch between the SiCs and the epoxy matrix. The mismatch is proof that phonons cannot easily be absorbed by SiCs’ crystalline structure during the transmission process, meaning the energy transferred by phonons is insensitive to the crystal lattice, which agrees with the earlier SEM results.

Meanwhile, the thermal diffusivity of the epoxy nanocomposites showed a similar change trend to the thermal conductivity, as shown in Figure 3b. According to Figure 3b, the thermal diffusivity of the samples showed a monotonic increase as more MWCNTs and GNPs were incorporated, but there was a slight decrease in the SiC nanofillers when the loading reached 4 vol.%. As shown in the figure, the thermal diffusivity of the pristine epoxy was around 0.164 mm^2^/s. With the addition of 1 vol.% of fillers, the thermal diffusivity of the epoxy/MWCNT 1, epoxy/GNP 1 and epoxy/SiCs 1 composites increased to 0.194 mm^2^/s, 0.181 mm^2^/,s and 0.172 mm^2^/s with about 18.26%, 10.30%, and 4.84% enhancement, respectively. As the content of the fillers further increased to 5 vol.%, the thermal diffusivity improved from 0.164 mm^2^/s to 0.315 mm^2^/s, 0.225 mm^2^/s, and 0.194 mm^2^/s for the epoxy/MWCNT 5, epoxy/GNPs 5, and epoxy/SiCs 5 with thermal diffusivity enhancing to 92.55%, 37.41%, and 18.48%, respectively. These data prove that the MWCNTs helped enhance the epoxy matrix’s thermal properties far beyond those resulting from the addition of the GNPs and SiCs.

The high aspect ratio of the graphene and CNTs may help to cause a significant increase in the composite’s thermal conductivity, enhanced by adding MWCNTs and GNPs [35]. Because CNTs (3000 W/mK) and graphene (5000 W/mK) have substantially higher intrinsic thermal conductivity than SiC (360 W/mK), nanoparticles containing a higher carbon nanofiller ratio have a greater ability to transmit heat [36]. In our study, the aspect ratio and lateral size of the SiCs were substantially smaller than those of the graphene and nanotubes, suggesting that there was a corresponding increase in filler–filler at the same filler-loading filler–epoxy interfacial interactions in the Epoxy/SiCs composites. As a result, the epoxy/SiC composite’s heat conduction efficiency was lower due to its high thermal boundary resistance [37]. This is supported by the SEM characterisation of the fracture surface morphologies of the composites in the next section.

The crosslinking density of epoxy and its composites is known to influence the glass transition temperature, *T_g_* [7]. Because the epoxy matrix has varying degrees of crosslinking, depending on the curing technique, the *T_g_* appears over a wider temperature range. Curing at room temperature results in lower *T_g_* values, while higher *T_g_* values necessitate higher-temperature curing. Therefore, the *T_g_* can be determined at the beginning, middle, or end temperatures. Herein, the intermediate temperature is defined as *T_g_*.

In this study, DSC was employed to compare the *T_g_* of the pure epoxy and epoxy nanocomposites loaded with 5 vol.% of fillers. Figure 4 shows the DSC thermograms as a function of temperature for the pure epoxy, epoxy/MWCNT 5, Epoxy/GNP 5 and epoxy/SiC 5 nanocomposites. The glass transition of the samples was investigated in the temperature range of 25–250 °C. From Figure 4, it can be observed that the *T_g_* of the pure epoxy was 61.09 °C, while the *T_g_* values of the epoxy/MWCNT 5, epoxy/GNP 5 and epoxy/SiCs 5were 87.94 °C, 91.67 °C, and 94.13 °C, respectively. It was proven that the epoxy had a low glass *T_g_* value compared with the other epoxy nanocomposites—the low *T_g_* of pure epoxy resin was attributed to specific movable and flexible epoxy polymeric networks. Hence, the *T_g_* results of the epoxy nanocomposites shown in this study were comparable with those of previously reported research investigations [38]. The *T_g_* of the epoxy/MWCNT 5 composit was enhanced by 26.85 °C. At the same loadings (5 vol.%), the *T_g_* values of the epoxy composites incorporated with GNPs and SiCs were enhanced at 30.58 °C and 33.04 °C, respectively. With the addition of the MWCNTs, GNPs, and SiCs to the epoxy matrix, the *T_g_* values increased, indicating that a robust interface was formed by the reaction of nanofillers with the matrix molecules during the curing process. This generated more obstacles to impede the macromolecular chain motion, resulting in greater *T_g_* values and enhanced thermal stability. The addition of nanofillers to the epoxy polymer acted as an obstruction and increased the variability of the crosslinked structure [38]. Variability can be promoted by minimising the voids and finally restricting the mobility of polymer within epoxy nanocomposites. The visible restriction on the movement of the polymeric network considerably increased the *T_g_* values of the epoxy nanocomposites. The obtained results were also validated by other researchers [39,40]. Incorporating nanofillers into the epoxy network reduced the polymeric chain movement either from chemical or physical interaction, improving the epoxy nanocomposites’ *T_g_.*

The thermal stabilities of the pure epoxy and epoxy nanocomposites were obtained using TGA analysis, as shown in Figure 5, where their behaviour was recorded from 50 °C to 750 °C. The TGA curves of the epoxy nanocomposites show a favourable effect of adding MWCNT GNP and SiC nanofillers to the epoxy matrix. The findings imply that the nanofillers placed in the epoxy matrix could limit the thermal motion of the polymer chains and the mobility of the polymer fragments at the epoxy interfaces. Based on Figure 5, three-stage decomposition levels can be observed in every sample condition. The first stage consisted of the thermal deterioration of all the epoxy nanocomposites and pure epoxy resin, with a loss of weight between 100 °C and 200 °C because of the elimination of physically adsorbed moisture via the dehydration of secondary functional groups and the evaporation of weak and loosely bound moisture from the composite surfaces. Similar trends have been reported in previous studies [38,41].

The second stage of decomposition occurred at approximately 320 °C, representing the decomposition of pure epoxy and epoxy in the matrix [38], as well as the removal of functional groups from the nanofillers, known as the pyrolysis of the carboxylic acid (–COOH) group from the surfaces of nanofillers [41]. The degradation of aromatic functional groups in the polymeric epoxy chain and the decay of aliphatic amine in the second stage were found during the decomposition of pure epoxy and epoxy nanocomposites [38]. The onset of the decomposition of the temperature of the pure epoxy was higher than that of the nanocomposites owing to the high thermal conductivity of the nanofillers (MWCNTs, GNPs, and SiCs), resulting in the rapid diffusion of heat in the matrix. Nanofillers enabled the matrix to transfer energy prior to decomposition, hence shortening the entire process. The epoxy matrix’s thermal conductivity was low, which slowed the rate of thermal degradation. The onset of the decomposition of the temperature of the pure epoxy was delayed by nearly 1 min compared with all the nanocomposites, according to the heating rate of 10 °C/min^−1^ determined in the thermogravimetric analysis. This observation is consistent with that of Guo et al. [42]. According to Wang et al., the oxygen-containing groups on MWCNT nanofillers thermally degrade at 250–350 °C, and the structure of MWCNT nanofillers thermally degrades at 450–800 °C [43]. In this region, more weight loss was observed for MWCNT nanofillers compared with those of the GNP and SiC nanofillers. Hydroxyl-functionalised MWCNT nanofillers consist of more carboxylic acid groups; hence, the deterioration was triggered by the breakdown of the oxygen-containing groups, resulting in more weight loss in the MWCNT nanofillers.

The third stage of decomposition was initiated at temperatures exceeding 400 °C, which involves the elimination of amorphous carbon. Wei et al. [41] obtained similar results, observing the elimination of amorphous carbon at temperatures exceeding 300 °C, while Moraitis et al. [44] discovered that amorphous carbon was removed at 500 °C. According to Figure 5, the functionalised MWCNT nanofillers had greater thermal stability than the unfunctionalised GNP nanofillers observed under temperatures ranging from 250 °C to 400 °C. However, at temperatures above 400 °C, the unfunctionalised GNP nanofillers outperformed the MWCNT nanofillers. This was due to additional carboxylic acid groups on the functionalised MWCNT nanofillers, accelerating the MWCNTs’ thermal decomposition.

At the last stage, at temperatures above 500 °C, the complete decomposition of the epoxy network was observed for the pure epoxy polymer, which tended to have nearly no remaining components. However, in the case of the epoxy nanocomposites reinforced with MWCNT, GNP, and SiC nanofillers, char was produced throughout under a high-temperature range. Due to greater crosslinking in the presence of nanofillers combined with homogeneous dispersion, the thermal degradation profile of the epoxy nanocomposites was improved by adding nanofillers. As a result, the thermal degradation was higher compared to that of the cured pure polymer [38]. The composite residue at 600 °C indicates that incorporating nanofillers into the epoxy matrix can promote the thermal stability of the epoxy itself [42].

### 3.3. Compression Properties

Compressive stress–strain curves for the pure epoxy and epoxy nanocomposites filled with MWCNTs, GNPs, and SiCs at a filler loading of 4 vol.% are shown in Figure 6. Because the compressive stress–-strain curves of all the samples are comparable, Figure 6 illustrates the compressive stress–strain curves of the pure epoxy and epoxy nanocomposites at selected filler loadings. As illustrated in Figure 6, the compressive curve can be categorised into four different regions. Region one is the initial elastic region (stress increasing linearly with strain), region two is the plastic deformation region (stress rising non-linearly with strain), region three is the plateau region (stress remaining relatively consistent with strain), and, the last region is the strain-hardening region (stress gradually increasing with strain due to the sample’s densification).

Figure 7 (compressive strength) and Figure 8 (compressive modulus) show the compression properties of the epoxy matrices filled with MWCNT, GNP, and SiC nanofillers as a function of filler loading (vol.%) produced using the planetary centrifugal method. Despite its low modulus (0.71 GPa), pure epoxy has a compressive strength of 54.26 MPa, indicating its good compliance. The variation of the compressive modulus of the Epoxy/MWCNTs, Epoxy/GNPs, and Epoxy/SiCs followed a similar trend to that of the compressive strength, indicating that the addition of nanofillers boosted the modulus and strength of the epoxy composite system up to a loading of only 4 vol.%. Regrettably, the subsequent additions at a loading of 5 vol.% diminished the compressive strength and modulus for all the epoxy composite systems with slightly decreased graphite-based fillers (MWCNTs and GNPs) and considerably decreased ceramic-based fillers (SiCs).

The compressive modulus of the epoxy/MWCNT 4 was 7.01× greater than that of the pure epoxy, which was 0.498 GPa. With 0.484 GPa, the compressive modulus of the epoxy/GNP 4 appeared to be the highest of the epoxy/GNP composites, demonstrating a 6.82× increment over the pure epoxy. On the other hand, the epoxy/SiC 4 demonstrated the greatest compressive modulus of all the epoxy/SiCs at 0.476 GPa, representing a 6.70× increase over the pure epoxy. Similarly, the compressive strength of the wpoxy/MWCNTs filled with 4 vol.% MWCNTs was found to be the highest, increasing by up to 192% (158.64 MPa) when compared to the pure epoxy. In the case of the epoxy/GNP composites, the compressive strength composite filled with 4 vol.% GNPs displayed the highest compressive strength, with an enhancement of up to 148% (134.50 MPa) above the pure epoxy. Finally, for the epoxy/SiCs, the compressive-strength composite filled with 4 vol.% SiCs demonstrated the highest compressive strength, increasing up to 96% (106.26 MPa) above the pure epoxy. As mentioned above, this indicates that the epoxy nanocomposites created using this method showed greater dispersion and distribution. However, further filler loadings greater than 4 vol.% compressive strength and modulus showed a decreasing trend because the nanofillers formed larger agglomerations in the epoxy matrices, making it easier for the overlapped tubes, sheets, and particles to slide past one another. This led to early crack initiation in the composites due to the nanofillers’ agglomeration, which created difficulty in dispersing the filler at higher loadings, particularly with the large surface area of the MWCNTs, GNPs, and SiCs. This is comparable to the report by Sheshkar et al. [16]. On the other hand, the compressive strength and modulus of the epoxy/MWCNTs were higher than those of the epoxy/GNPs and epoxy/SiCs because a better filler–matrix interaction can create a stronger reinforcement effect than that of a poor filler–matrix interaction, which has more stress-cracking points. As a result, the MWCNT nanofillers could create additional interfaces between one another and the resin. These interfaces may interact during the deformation process, increasing the composite’s strength while reducing the compressive strain. This is comparable to the report by Raza et al. [45]. Another study, by Naeimirad et al. [24], demonstrated that the flexural and impact strength of the epoxy/SiC composites initially improved with the addition of SiC nanoparticles but deteriorated with an excessive quantity of SiCs. Hemath and Selvan [38] found that every epoxy nanocomposite sample failed more forcefully and in a dissimilar manner owing to the agglomeration of a large number of nanoparticles, which led to the propagation of micro-cracks on the surface of the material.

The compressive strength of the epoxy/SiCs 5 nanocomposites demonstrated a considerable decrease while the compressive modulus slightly reduced compared with the epoxy/MWCNT 5 and epoxy/GNP 5 nanocomposites. This also proved that SiCs, as ceramic-based fillers, can produce stiffer composites than graphite-based fillers at equivalent filler loadings. The surface area of the SiCs used in this study was lower than those of the MWCNTs and GNPs. Thus, the interfacial area of the SiCs with the epoxy was lower than that of the MWCNTs and GNPs with the epoxy matrix. The lower interfacial contact area for the former meant that the SiC nanofillers could develop much more extensive interconnections than those between the graphite-based fillers due to their nanotubes and platelet morphology. This is discussed in more detail in the following section (on the fracture surface morphology). Consequently, there was less interstitial resin to interfere with the SiC particle interactions, thereby allowing more interactions between the SiC particles under compression. When the epoxy/SiCs 5 nanocomposites lost strength, they were less compliant. Meanwhile, the higher interfacial area between the MWCNT and GNP nanofillers and resins and the greater dispersion of MWCNTs and GNPs on a sub-microscopic level allowed fewer interconnections between the fillers. Therefore, less interconnectivity was required between the nanofillers because the interfacial area between the MWCNT and GNP nanofillers was larger, and the MWCNTs and GNPs were dispersed more evenly. The epoxy could be deformed before the stress was transferred to the fillers, resulting in composites with improved strength and compliance. This comparison suggests that the graphite-based fillers (MWCNTs and GNPs) produced composites with a higher thermal conductivity than those prepared using the ceramic-based filler composites at significantly lower filler loadings, producing more compliant composites than in the latter case. Epoxy resin reinforced with graphite-based and ceramic-based fillers has similar compression strength behaviour, as reported by Hemath and Selvan [38] and Raza et al. [46].

### 3.4. Fracture Surface Morphology

The fracture surfaces of the pure epoxy and epoxy nanocomposites were evaluated using SEM, as shown in Figure 9, Figure 10, Figure 11 and Figure 12, to understand the distribution and dispersibility of the nanofillers in all the nanocomposites and to establish the relationship between the structure and its properties. Figure 9 shows the fracture surface morphology of the pure epoxy, while Figure 9, Figure 10, Figure 11 and Figure 12 show the fracture surface morphology of the epoxy nanocomposites filled with the lowest filler loading (1 vol.%) and the highest filler loading (5 vol.%) of MWCNT, GNP, and SiC nanofillers, respectively. Figure 9, Figure 10, Figure 11 and Figure 12 demonstrate striped structures with fractures, which feature river-like patterns, and these patterns are similar to the fracture surface patterns in the pure epoxy shown in Figure 9. According to the SEM images of the fracture surface in the pure epoxy sample (Figure 9), the hackle bands and crack advances seen in the pure epoxy sample are the typical features of a cleavage brittle fracture in most amorphous polymers. The very smooth fracture surface it indicates that the composite is a brittle thermosetting polymer. Similar observations have been reported by Wang et al. [32], Hoseini et al. [47], and Shen et al. [48]. Figure 9, Figure 10, Figure 11 and Figure 12 display that the epoxy composites’ fracture surfaces had distinct fracture graphic features after adding the nanofillers.

Figure 10 shows the fracture surfaces of the epoxy/MWCNTs after the compression failure. Almost no naked nanotubes can be seen in the smooth cross-sectional pattern in Figure 10a,b, which shows a homogeneously dispersed distribution of MWCNTs in epoxy/MWCNT 1. A crosslinked nanotube network and high interfacial interaction between the epoxy matrices and the surfaces of the nanotubes are thought to be the critical factors in the composite’s thermal characteristics because of the use of hydroxyl-functionalised MWCNTs in this study. A local network of CNTs could be formed by 5 vol.% MWCNTs, and a 3D thermal conductivity network could be constructed inside the composite material. However, it was difficult to obtain a uniform dispersion of CNTs in epoxy nanocomposites at higher filler loadings. Fewer large agglomerations of nanotubes were observed in the epoxy/MWCNT 1 and epoxy/MWCNT 5, and it was quite challenging to disperse them in the epoxy matrix. This observation is in accordance with the results obtained by Wladyka-Przybylak et al. [49]. The SEM images presented in Figure 10c,d show that the MWCNTs in the epoxy/MWCNT 5 composites failed to disperse individually, and fewer agglomerated nanotubes were formed. The dispersion of individual MWCNTs is very difficult owing to their tendency to form large agglomerations because of the intermolecular van der Waals interactions between the individual nanotubes. Nanocomposites with high dispersion are preferred to provide better thermal conductivity, as the agglomeration of nanocomposites can reduce their usefulness as heat conductors [50]. Analysing MWCNTs’ dispersion is complicated because the dispersion changes depending on the magnification used in the analysis. CNTs’ dispersion can be classified into two aspects: the loosening of CNT bundles or agglomerates, which is a nanoscopic dispersion; and the uniform spreading of individual CNTs or their agglomerates throughout the nanocomposites, which is more of a micro- and macroscopic dispersion, as defined by Li et al. [51].

Therefore, optical microscopy and Raman imaging analyses can be used to determine the regularity of the agglomeration and distribution of MWCNTs, while SEM images can reveal whether the MWCNTs are disentangled, as suggested by Gardea et al. [50]. The MWCNTs’ dispersion is described qualitatively at an observational length scale. From Figure 10a–d, it can be seen that the surface irregularity of the nanocomposites was also enhanced as an epoxy-rich matrix became an MWCNT-rich matrix (1–5 vol.% of filler loading) and was well dispersed, with no dragging out of MWCNTs, indicating that the MWCNTs were well dispersed within the epoxy resin. The SEM image in Figure 10d shows that the MWCNTs were pulled out from the crack-oriented area and underwent debonding. Ali et al. [52] have also demonstrated this in their research. The fracture shows that the MWCNTs and epoxy had strong adhesion and interfacial contact between them. The MWCNTs had a stiff structure; therefore, cracks appeared randomly. Compared with the pure epoxy, the addition of MWCNTs into the epoxy matrix caused an increase in the surface roughness of the fracture surface. The SEM dispersion analysis shows that the epoxy/MWCNTs samples had smaller agglomerates, with a greater distance between them, compared to those of the epoxy/GNPs and epoxy/SiC nanocomposites. This was also reported by Gardea et al. [50].

Figure 11 shows the fracture surface morphology of the epoxy/GNP nanocomposites. As shown in Figure 11, the GNPs were uniformly disseminated in the epoxy composite with no visible aggregates and tightly encased in the epoxy polymer, which suggests strong interfacial contacts in the composites. The epoxy/GNP nanocomposites also exhibited river patterns on the fracture surface that were nearly identical to those found in the pure epoxy. The epoxy/GNP nanocomposites, in contrast to the epoxy/MWCNT nanocomposites, had some cracks on the fracture surface, as seen in Figure 11a–d. The epoxy’s fracture surface became rougher and more distorted as the loading of the GNPs increased, as illustrated by the numerous convoluted indentations and deep cracks depicted in Figure 11d. Unlike the epoxy filled with MWCNTs, in the matrix filled with GNPs, the GNPs protruded cleanly from the fracture surface, indicating a weak interfacial interaction between the epoxy matrix and the GNPs [35]. The weak bonding between the filler and the matrix is seen in Figure 11b, where a space between the filler platelets and the matrix was observed between the individual platelets. This void resulted in heat flow restriction, which inhibited the phonon transport channels, reducing the nanocomposite’s thermal conductivity. Figure 11 depicts the fracture surfaces, and the fracture followed the path of least resistance, where there was weak filler-to-matrix adhesion. Comparing the MWCNTs and GNPs, as both of them are graphite-based fillers, the SEM images of the epoxy/MWCNTs in Figure 10 show a nanocomposite packed with agglomerates of MWCNTs, which had a considerably higher degree of continuity between the particles and the matrix, as well as between individual particles. Because the agglomerates were near one another, and the particle–particle contact zone was embedded in an epoxy matrix coating, these two components worked together to generate a continuous phonon transport channel. As a result, the MWCNT agglomerates had a higher thermal conductivity than the GNP platelets [53].

Figure 12 shows the fracture surface morphology of the epoxy/SiCs nanocomposites. Unlike the epoxy/MWCNT and epoxy/GNP nanocomposites, the microstructures of the epoxy/SiCs nanocomposites were not perfect, as shown in Figure 12a–d. There were large voids between the filler agglomerations. These voids considerably lowered the thermal conductivity, and their presence was determined by measuring the density of the composites, as discussed above. This void content must be eliminated or reduced in epoxy/SiC nanocomposites if greater thermal conductivities are to be achieved. Using pressure during curing is one option that is currently being researched. Compared with the dispersion states of the nanofillers used in this study, the epoxy/SiC nanocomposites exhibited more aggregation of SiC particles due to their poor dispersion stability. This implies that the increase in the SiC volume ratio in the composites promoted the poor dispersion of the SiCs, facilitating the reduction in the thermal conductivity of the composites. This finding is similar to that reported by Ren et al. [7]. Without any exception of filler loadings in the epoxy/SiCs nanocomposites, all the materials showed varying degrees of agglomeration, which is an inevitable problem with nanocomposites. As demonstrated by the fracture surface shown in Figure 12c,d, the 5 vol.% SiCs could not form an effective network of SiC nanoparticles, and the fillers were isolated from each other in the epoxy matrix. The presence of nanoparticles resulted in small fracture planes. Due to the uniform dispersion of the nanoparticles in the epoxy matrix, the cracks could only move a short distance before being intercepted by an individual nanoparticle.

However, at the fracture surface of the epoxy/SiC 5, the agglomeration of nanoparticles and the formation of air voids were visible, as can be seen in Figure 12c,d. Figure 11a,b shows that an approximate dispersion was obtained for the nanofiller loadings of 1 vol.% SiCs, whereas for the loadings of 5 vol.%, significant agglomeration occurred in the composites. In the high loadings of SiC nanofillers, agglomeration was caused by the interaction of the fillers; a similar observation was made by Hoseini et al. [47]. The morphology analysis of the fracture surface using SEM revealed that agglomerates formed in the epoxy nanocomposite samples, particularly SiCs, and especially at high filler loadings. Thus, the slight increase in thermal conductivity could have be caused by the presence of agglomerated nanofillers. Gardea et al. [50] also mentioned that agglomerated nanofillers could immensely affect the thermal conductivities of composites. Therefore, if a major improvement in thermal conductivity is the main requirement, the individual dispersion of the nanofillers in the matrix becomes an extremely crucial factor to address.

## 4. Conclusions

Thermally conductive epoxy composites have been utilised in various electronic applications, such as thermal interface materials, electronic packaging, and energy storage devices. The effects of different nanofillers on epoxy nanocomposites’ mechanical and thermal properties were investigated. The graphite-based fillers, MWCNTs and GNPs, were more effective reinforcements for the epoxies concerning mechanical and thermal conductivity than the ceramic-based fillers, SiCs. However, these inorganic fillers often became sedimented during the curing reaction of the epoxy matrix. The relatively uniform dispersion of the MWCNT and GNP nanofillers was achieved using ultrasonication combined with planetary centrifugal mixing, but this was not the case for the SiC nanofillers, where significant agglomeration was observed in each of the filler loadings. This was possible in the epoxy/SiCs at very low loadings because of the nanoparticles’ highly dispersive surface energy, which produced spontaneous re-agglomeration. The following conclusions can be drawn from the study’s findings:Among these nanofiller materials, the MWCNTs showed the highest compressive strength and modulus, followed by up to 4 vol.% filler loadings of GNPs and SiCs, with an increase of 266%, 240%, and 224%, respectively, compared to the pure epoxy. This was because the MWCNTs prepared had a greater dispersion and distribution in the epoxy matrix. Since the MWCNTs were hydroxyl-functionalised MWCNTs, the chemical modification improved the interfacial adhesion through amine linkages; therefore, the MWCNTs fillers could create additional interfaces between the filler-to-filler and filler-to-resin, resulting in a better filler–matrix interaction and enhancing the stress transfer from the epoxy matrix to the MWCNT nanofillers. However, all the nanocomposites above 4 vol.% filler loadings showed a decreasing trend in compressive strength and modulus because many agglomerations occurred in the composites, which became the stress-cracking points of the composites.The thermal stability of the MWCNTs was slightly lower than those of the GNPs and SiCs. The low thermal stability of the functionalised MWCNTs was attributed to the oxygen-containing groups. Despite this, the thermal conductivity of the epoxy/MWCNTs was higher than those of the Epoxy/GNPs and Epoxy/SiCs. Furthermore, the SEM analysis revealed that the fillers were close to each other in the epoxy matrix, forming a promising pathway for phonons to travel in the epoxy matrix. Thus, less scattering of phonons occurred, and more phonons could move at a faster transmission rate through the conductive pathways, leading to an increase in thermal conductivity.The morphological analyses indicated that the nanocomposites failed under compression loading in typical modes, such as filler pull-out, debonding, and matrix cracking. For most of the nanocomposites, the stress-cracking points were initiated by the agglomeration of the nanofillers, propagated through the huge agglomerated fillers and, lastly, transferred to the void-filled epoxy matrix.

## Figures and Tables

**Figure 1 polymers-14-01014-f001:**
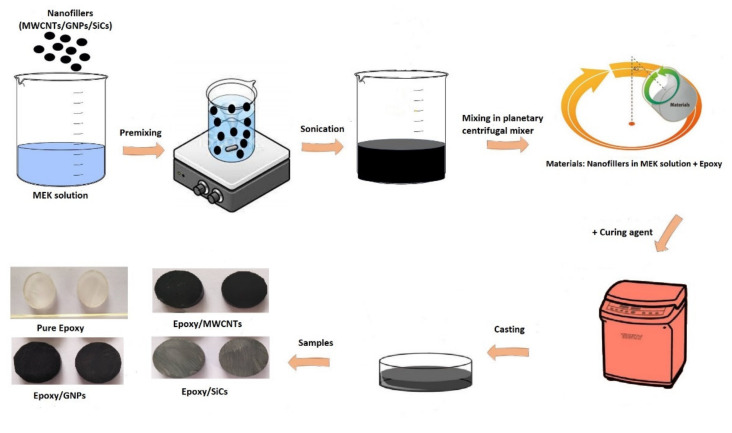
Illustration of the preparation process of the epoxy nanocomposites.

**Figure 2 polymers-14-01014-f002:**
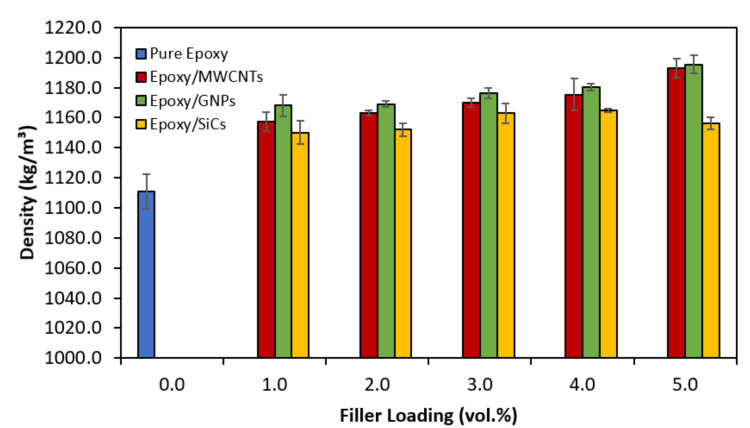
Density as a function of vol.% for epoxy nanocomposites filled with MWCNTs, GNPs and SiCs.

**Figure 3 polymers-14-01014-f003:**
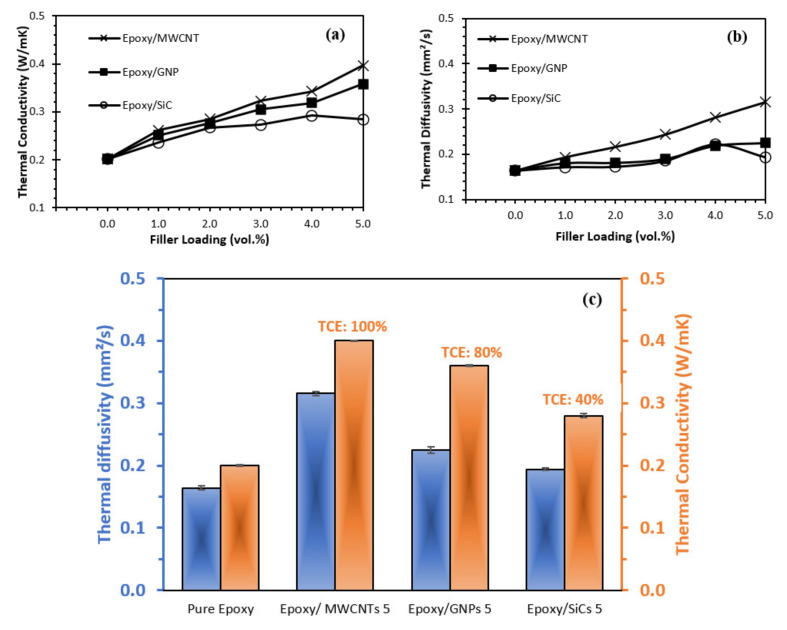
(**a**) Thermal conductivity, (**b**) thermal diffusivity of epoxy nanocomposites at various filler loadings (1–5 vol.%), and (**c**) thermal conductivity and diffusivity enhancement of epoxy nanocomposites at 5 vol.% filler loading.

**Figure 4 polymers-14-01014-f004:**
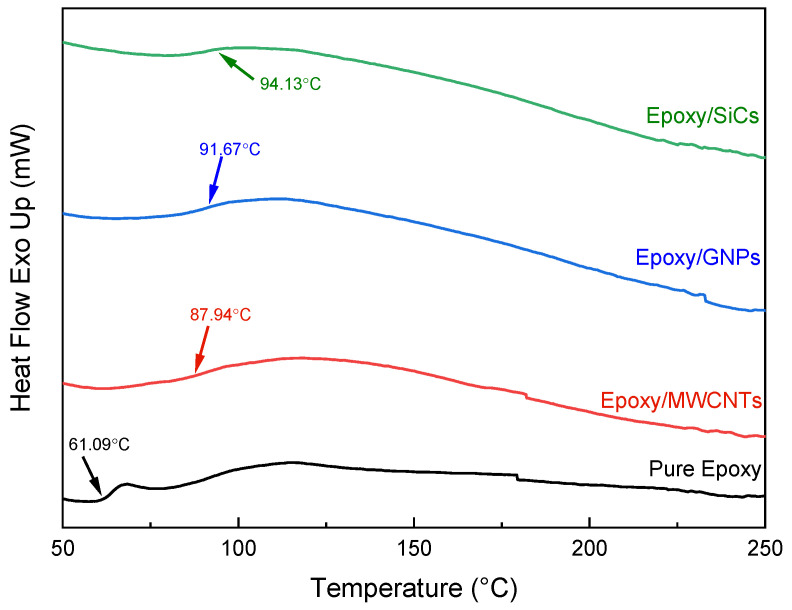
DSC curves of pure epoxy and epoxy nanocomposites filled with MWCNTs, GNPs, and SiCs.

**Figure 5 polymers-14-01014-f005:**
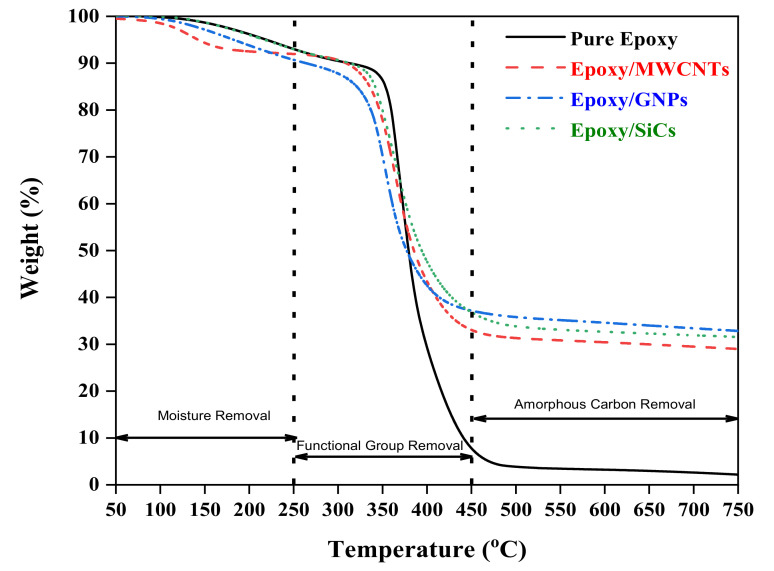
TGA curves of pure epoxy and epoxy nanocomposites filled with MWCNTs, GNPs, and SiCs.

**Figure 6 polymers-14-01014-f006:**
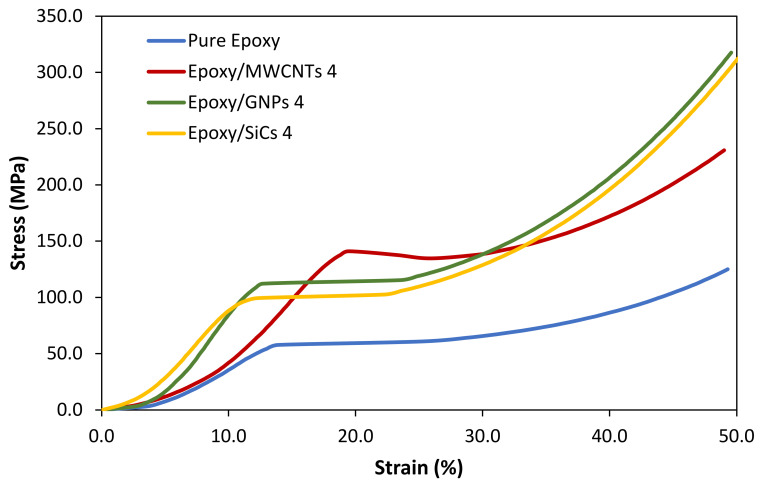
Typical compression stress–strain curves of pure epoxy and epoxy nanocomposites filled with MWCNTs, GNPs, and SiCs at 4 vol.% filler loading.

**Figure 7 polymers-14-01014-f007:**
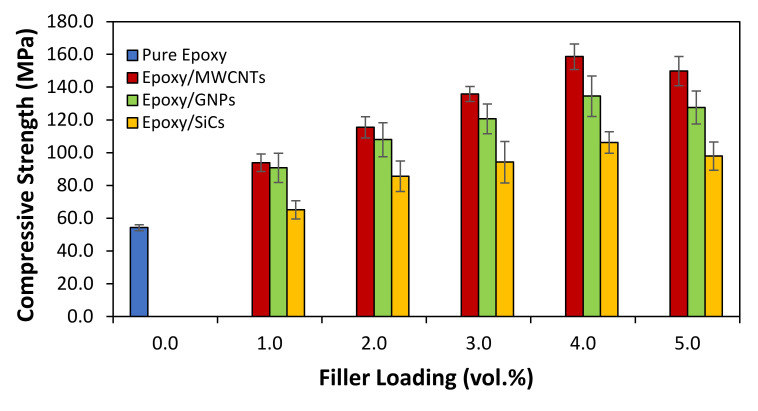
Compressive strength of pure epoxy and epoxy nanocomposites filled with MWCNTs, GNPs, and SiCs.

**Figure 8 polymers-14-01014-f008:**
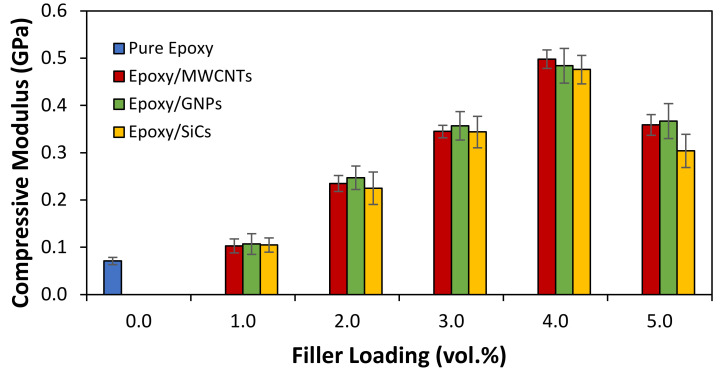
Compressive modulus of pure epoxy and epoxy nanocomposites filled with MWCNTs, GNPs, and SiCs.

**Figure 9 polymers-14-01014-f009:**
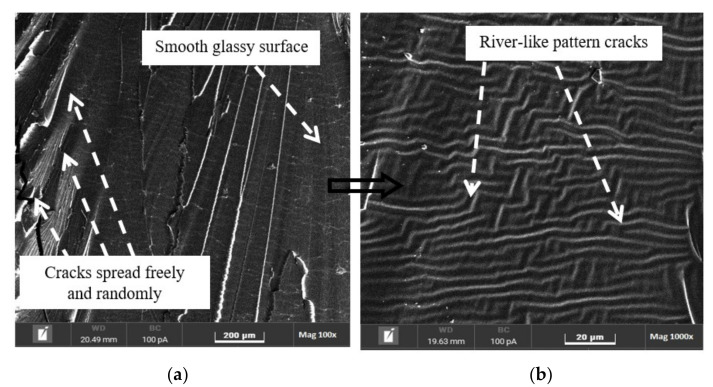
Scanning electron microscopy images of the fracture surfaces of pure epoxy at (**a**) 100× magnification and (**b**) 1000× magnification.

**Figure 10 polymers-14-01014-f010:**
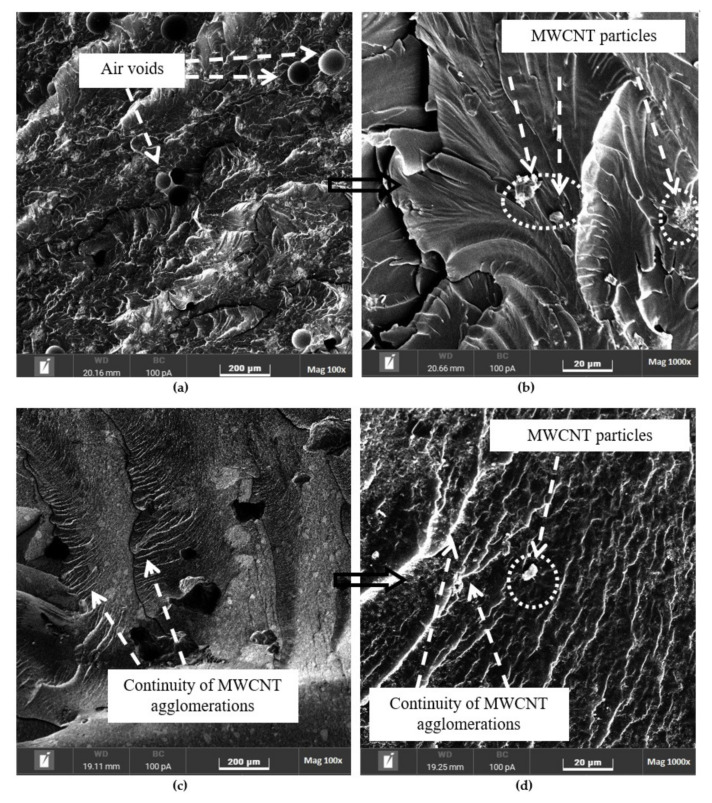
Scanning electron microscopy images of the fracture surfaces of (**a**,**b**) 1 vol.% and (**c**,**d**) 5 vol.% epoxy/MWCNT nanocomposites.

**Figure 11 polymers-14-01014-f011:**
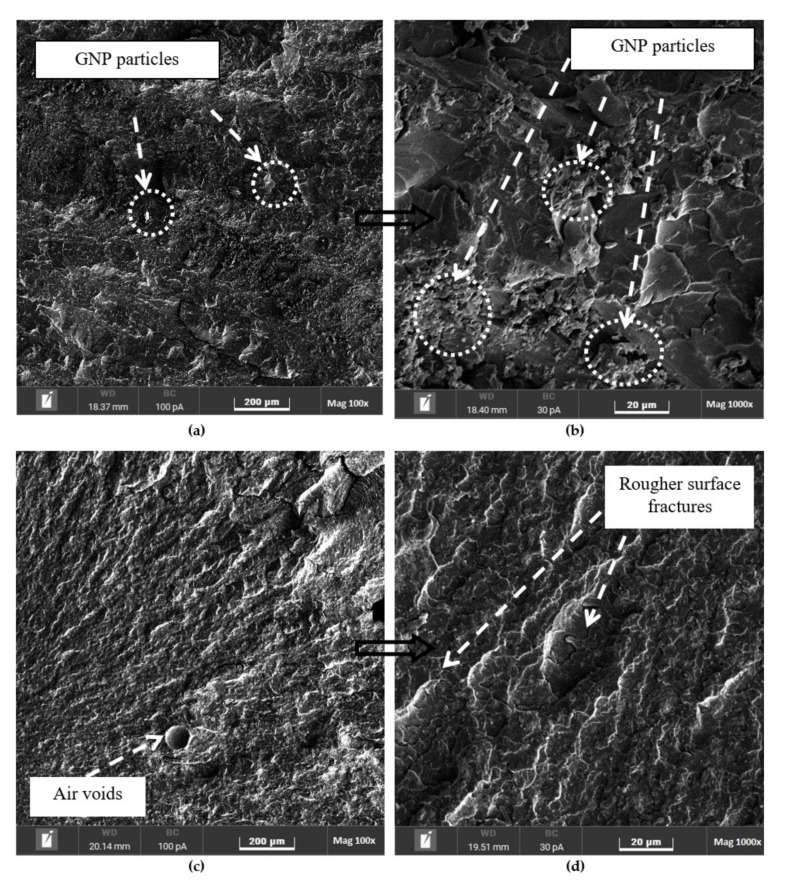
Scanning electron microscopy images of the fracture surfaces of (**a**,**b**) 1 vol.% and (**c**,**d**) 5 vol.% epoxy/GNP nanocomposites.

**Figure 12 polymers-14-01014-f012:**
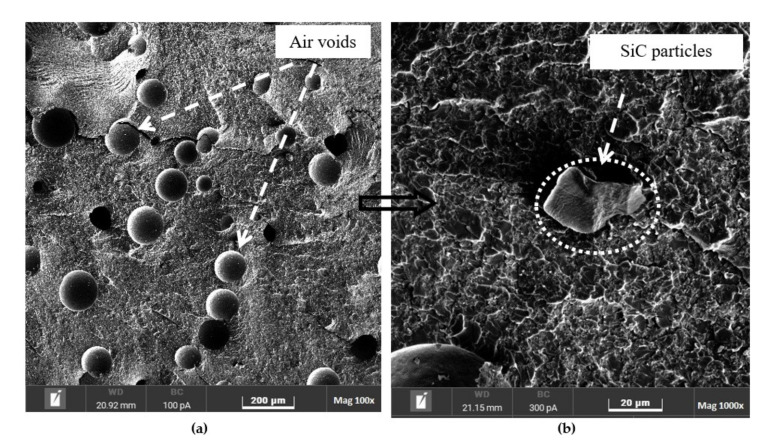
Scanning electron microscopy images of the fracture surfaces of (**a**,**b**) 1 vol.% and (**c**,**d**) 5 vol.% epoxy/SiC nanocomposites.

**Table 1 polymers-14-01014-t001:** Properties of fillers.

**Filler**	**MWCNTs**	**GNPs**	**SiCs**
Appearance	Black powder	Black powder	Greyish powder
Morphology	Nanotubes	Platelet	Cubic
Content of carbon (%)	>97	>99	-
Average diameter/particle size (µm)	20.0–40.0 nm	<2.0 µm	40 nm
Length (µm)	<10.0	<2.0	2.0–60.0
Thickness average (nm)	0.8–1.6	1.0–5.0	45.0–65.0
Density given by supplier (g/cm^3^)	2.1–3.0	2.0–2.2	3.217
Density calculated by PDA (g/cm^3^)	2.0783	2.1080	3.3004
Surface area given by supplier (m^2^/g)	210.0–300.0	750.0	30.0–60.0
Surface area by BET (m^2^/g)	236.9138	770.3366	35.8165
Thermal conductivity (W/mK)	3000	5000	360
Electrical conductivity (S/m)	10^7^	10^7^	-

## Data Availability

There are no linked research datasets for this submission. Data will be made available on request.

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
