# Peer review of "Physical, Thermal Transport, and Compressive Properties of Epoxy Composite Filled with Graphitic- and Ceramic-Based Thermally Conductive Nanofillers"

_polymers, 2022, doi:10.3390/polym14051014_

Round 1

Reviewer 1 Report

Useful work.

1 The author should specify the scope of application. How high thermal conductivity is needed.

2 Usually, the excellent electrical insulation  is one the ,ost important performances for electronic packaging. How about the properties?

3 Is there an optimized curing process with different filler doping?

Reviewer 2 Report

Review report on ‘Physical, Thermal Transport and Compressive Properties of Epoxy Composite Filled with Graphitic- and Ceramic-Based Thermally Conductive Nanofillers’. The detailed comments are listed below:

  1. The abstract section is very lengthy. Remove the unnecessary information and add the key finding of the works.
  2. The introduction section is very weak. Add more references related to the current work and define the gap clearly. Also add separate paragraph indicating the novelty of the work.
  3. The experimental setup is missing. Add the images of the samples.
  4. Add more detail about the compression testing measurement along with the ASTM standard used for sample preparation.
  5. Add the compression test stress-strain plots.
  6. The compressive strength of the nanocomposites decreases after 4 % volume filler. The technical discussion related to strength variation is missing.
  7. Can authors review the theoretical modeling of the Epoxy Composite Filled with nanoparticles?
  8. Add the following recent articles in the reference list.
  • https://doi.org/10.1016/j.matpr.2018.06.526
  • https://doi.org/10.1007/s10965-021-02774-w
  • https://doi.org/10.1007/s10965-020-02359-z

Round 2

Reviewer 2 Report

The authors have reacted adequately to my objections and fixed the pointed issues. The revised manuscript can be accept to be published.

Author Response

Please refer response attached
